# Palliative Radiotherapy in Metastatic Breast Cancer Patients on CDK4/6 Inhibitors: Safety Analysis

**DOI:** 10.3390/cancers17030424

**Published:** 2025-01-27

**Authors:** Furkan Ceylan, Mirmehdi Mehdiyev, Burak Bilgin, Ateş Kutay Tenekeci, Bülent Yalçın, M. Bülent Akıncı, Didem Şener Dede, Mehmet Ali Nahit Şendur, Efnan Algın, Şebnem Yücel

**Affiliations:** 1Department of Medical Oncology, Ankara Bilkent City Hospital, 06800 Ankara, Turkey; 2Department of Medical Oncology, Ankara Yıldırım Beyazıt University, 06800 Ankara, Turkey; 3School of Medicine, Hacettepe University, 06230 Ankara, Turkey

**Keywords:** CDK4/6 inhibitor, radiotherapy, hematologic toxicity, safety

## Abstract

This retrospective study evaluates the safety of combining CDK4/6 inhibitors with palliative radiotherapy in metastatic hormone receptor-positive, HER2-negative breast cancer. CDK4/6 inhibitors are essential in managing this disease but are associated with hematologic toxicities. Palliative radiotherapy, often used for bone metastases, shares similar side effects, raising concerns about their concurrent use. Involving 188 patients, the study revealed no significant increase in grade ≥ 3 hematologic toxicities, such as anemia or neutropenia, among patients receiving both treatments compared to those treated with CDK4/6 inhibitors alone. Ribociclib was linked to lower rates of severe hematologic toxicity, neutropenia, and fewer treatment interruptions and dose reductions compared to palbociclib. These findings suggest that palliative radiotherapy can be safely integrated with CDK4/6 inhibitors, providing critical insights into optimizing treatment strategies for patients with metastatic breast cancer. The study highlight the need for further prospective trials to confirm these observations and refine clinical guidelines.

## 1. Introduction

The use of cyclin-dependent kinase 4 and 6 inhibitors (CDK4/6 inhibitors) (palbociclib, ribociclib, and abemaciclib) in combination with endocrine therapy is the current standard of care for the first- or second-line treatment of hormone receptor-positive, HER2-negative metastatic breast cancer [1,2,3]; also, abemaciclib and ribociclib plus endocrine therapy have demonstrated an improvement in invasive disease-free survival in patients with HR+/HER2- early breast cancer whose tumors had a high risk of recurrence [4,5].

The widespread use of CDK4/6 inhibitors highlights the critical need to effectively understand and manage their associated side effects. The toxicity profiles of CDK4/6 inhibitors (palbociclib, ribociclib, and abemaciclib) can differ based on their selective inhibition of CDK4 and CDK6 [6]. Palbociclib exhibits similar selectivity for CDK4 and CDK6, leading to a predominantly hematologic toxicity profile. Ribociclib, with a slightly higher selectivity for CDK4 than palbociclib, shares a similar toxicity profile. In contrast, abemaciclib, which has a much more potent inhibition of CDK4 than CDK6, is associated with a higher incidence of non-hematologic toxicities, such as gastrointestinal side effects, and a comparatively lower incidence of hematologic toxicity [7,8]. Fatigue, nausea, liver function abnormalities, skin toxicity, and thromboembolism are other commonly observed adverse effects.

Radiotherapy can be used in early-stage disease to prevent local recurrence after surgery, as well as in the metastatic stage for palliative purposes. The risk of cytopenia increases when radiotherapy is applied to bone structures where hematopoiesis is commonly observed. There are conflicting results in studies evaluating whether radiotherapy increases hematologic toxicity when using CDK4/6 inhibitors [9,10,11]. These studies are retrospective and involve a small number of patients. Although interruption of palbociclib during radiotherapy is recommended in the PALOMA trial, no prospective randomized trials evaluate the safety of combined use of radiotherapy and CDK4/6 inhibitors.

We aimed to evaluate the impact of palliative radiotherapy on the adverse effect profile of CDK4/6 inhibitors.

## 2. Patients and Methods

### 2.1. Patient Selection

Patients with hormone-positive HER2-negative metastatic breast cancer who received CDK4/6 inhibitors in Ankara Bilkent City Hospital between January 2021 and June 2024 were retrospectively reviewed. Patients whose radiotherapy status was unknown were not included in the study. The study was designed as a retrospective, single-center study. Data on demographic and clinical characteristics, progression-free survival (PFS), and overall survival (OS) were extracted from medical records. Follow-up duration was calculated using the reverse Kaplan–Meier method. The primary endpoint was the incidence of grade ≥ 3 hematologic toxicity associated with palliative radiotherapy in patients receiving CDK 4/6 inhibitors. Secondary endpoints included identifying factors influencing the occurrence of grade ≥ 3 hematologic and non-hematologic toxicities, such as liver function abnormalities and acute kidney injury, as well as PFS and OS. The study focused on palliative radiotherapy administered to bone structures for pain relief and fracture prevention. Extensive bone metastasis was defined as 5 or more bone metastases.

### 2.2. Evaluation of Adverse Effects

Toxicity was evaluated using the National Cancer Institute Common Terminology Criteria for Adverse Events (CTCAE), version 5.0, for patients undergoing treatment with CDK4/6 inhibitors. Blood samples collected throughout the treatment were analyzed, and examination findings, hemograms, and biochemistry test results were meticulously recorded. The highest recorded values were used for the toxicity assessment. An interruption in CDK4/6 inhibitor therapy was defined as temporarily discontinuing the medication until the subsequent scheduled follow-up. Dose reduction was characterized as decreasing the dosage to 400 mg/day or 200 mg/day for ribociclib and 100 mg/day or 75 mg/day for palbociclib. Withdrawal refers to the permanent cessation of the medication. PFS was defined as the time from the initiation of CDK4/6 inhibitor therapy to either disease progression, the last recorded follow-up, or death. OS was defined as the time from the initiation of CDK4/6 inhibitor therapy to death or the last recorded follow-up.

### 2.3. Statistics

Descriptive statistics were reported as mean with standard deviation or median with interquartile range, depending on the distribution of the variables. For comparisons of numerical variables, Student’s *t*-test was employed for normally distributed data, while the Mann–Whitney U test was utilized for non-normally distributed data. The Chi-Square test was used to compare proportions in categorical variables. Progression-free survival (PFS) and overall survival (OS) were analyzed using the Kaplan–Meier method, with group comparisons made using the log-rank test. Cox regression analysis was performed to identify independent predictors of survival, and logistic regression analysis was used to determine independent predictors of hematological toxicity, drug interruption, and withdrawal. All statistical analyses were conducted using SPSS version 26.0 (SPSS Inc., Chicago, IL, USA). A *p*-value of less than 0.05 was considered statistically significant.

This study received approval from the institutional ethics review board of Ankara Bilkent City Hospital and was conducted according to the principles of the Declaration of Helsinki.

## 3. Results

### Patient and Tumor Characteristics

The study included 188 patients, with a median follow-up time of 18.5 months. The 18-month PFS and OS rates were 67% and 85%, respectively. Detailed patient and tumor characteristics are presented in Table 1. The cohort’s median age was 57.5 years, with 79% of patients being post-menopausal. Most patients had a PS of 0/1. The most common sites of metastasis were bone (66%), liver (23%), and lymph nodes (22%). Extensive bone metastases were observed in 46% of patients, and vertebral bone metastases were observed in 47% of patients. The rate of de novo metastasis was 63%. Among those treated with CDK4/6 inhibitors, 76% received them as first-line therapy, while 22% were treated in the second line. Aromatase inhibitors were used in 70% of patients and fulvestrant in 30%. Concurrent radiotherapy and CDK4/6 inhibitor therapy were administered to 25% of patients (*n* = 47). Patient characteristics were generally comparable between those who received radiotherapy and those who did not, with the exception of a higher rate of de novo metastases (70% vs. 46%, *p* = 0.003) and first-line CDK4/6 inhibitor use (82% vs. 57%, *p* = 0.006) in the non-radiotherapy group (Appendix A).

Of the 47 patients receiving concomitant radiotherapy, 44 underwent treatment with 20Gy delivered in 5 fractions, while 3 patients received 30Gy in 10 fractions. Regarding the timing of radiotherapy relative to CDK4/6 inhibitor therapy, 32 patients received radiotherapy within the first 3 months of treatment, 6 patients during the second 3 months, and 9 patients after 6 months. The choice of fractionation and dose was determined by the treating clinician and was guided by standard palliative radiotherapy guidelines [12]. A total of 30Gy was administered to the sternum and lumbar vertebrae. Palliative radiotherapy was administered to a single bone metastatic site per patient, with the treatment site selected based on clinical assessment of the most symptomatic or high-risk lesion requiring intervention. The most common sites treated with radiotherapy were the spine (29), femur (9), pelvis (4), sternum (2), humerus (2), and skull (1), respectively. Twenty-three patients (12%) had received chemotherapy within one year before initiating CDK4/6 inhibitors.

Kaplan–Meier survival curves comparing progression-free survival and overall survival between patients receiving CDK4/6 inhibitors with concomitant palliative radiotherapy and those receiving CDK4/6 inhibitors alone were included and analyzed in Appendix A. PFS and OS were comparable between groups.

Grade ≥ 1 hematologic toxicity was reported in 92% of patients receiving CDK4/6 inhibitors. The incidences of anemia, neutropenia, and thrombocytopenia were 65%, 90%, and 23%, respectively. Grade ≥ 3 hematologic toxicity occurred in 52% of patients, with the rates of grade ≥ 3 anemia, neutropenia, and thrombocytopenia being 10%, 47%, and 7%, respectively. Elevated ALT levels were observed in 23% of patients, with a 1% incidence of grade ≥ 3 ALT elevation. Additionally, grade 2 QT prolongation was reported in two patients, and grade 3 dermatitis was observed in one patient.

Table 2 summarizes the frequency of adverse effects according to treatment, patient, and tumor characteristics. The frequency of grade ≥ 3 hematologic toxicity (52% vs. 47%, *p* = 0.677) and grade ≥ 3 neutropenia (48% vs. 47%, *p* = 0.985) was comparable between patients with ECOG PS 0/1 and those with ECOG PS 2/3. Patients receiving fulvestrant experienced a higher incidence of grade ≥ 3 hematologic adverse events (59% vs. 48%, *p* = 0.174) and grade ≥ 3 neutropenia (57% vs. 43%, *p* = 0.071) compared to those receiving aromatase inhibitors (AIs). In patients aged 60 and older, the frequency of grade ≥ 3 hematologic adverse events (58% vs. 46%, *p* = 0.097) and grade ≥ 3 neutropenia (54% vs. 42%, *p* = 0.124) was higher compared to those under 60. Patients receiving palbociclib exhibited significantly more grade ≥ 3 hematologic adverse events (68% vs. 42%, *p* < 0.001), grade ≥ 3 anemia (16% vs. 7%, *p* = 0.056), and grade ≥ 3 neutropenia (61% vs. 39%, *p* = 0.005) compared to those receiving palbociclib. Furthermore, patients treated with CDK4/6 inhibitors in the first-line setting had fewer grade ≥ 3 hematologic adverse events (47% vs. 67%, *p* = 0.020), grade ≥ 3 anemia (8% vs. 16%, *p* = 0.167), and grade ≥ 3 neutropenia (43% vs. 60%, *p* = 0.051) compared to those receiving these inhibitors in subsequent lines of therapy.

Treatment interruption, dose reduction, and discontinuation rates were 55%, 24%, and 2%, respectively. There was no significant difference in the rates of CDK4/6 inhibitor interruption (53% vs. 56%, *p* = 0.735) and dose reduction (28% vs. 23%, *p* = 0.490) between patients who received palliative radiotherapy and those who did not.

Detailed results regarding drug interruption, dose reduction, and discontinuation are provided in Table 3. Drug interruption was less frequent among patients using aromatase inhibitors compared to those on fulvestrant (51% vs. 64%, *p* = 0.098), younger patients under 60 compared to those 60 years and older (48% vs. 64%, *p* = 0.026), those treated with ribociclib versus palbociclib (44% vs. 73%, *p* < 0.001), and those receiving first-line treatment compared to those on second or subsequent lines of therapy (53% vs. 67%, *p* = 0.079).

Dose reduction was less common in patients treated with ribociclib compared to palbociclib (16% vs. 37%, *p* = 0.001), in younger patients under 60 compared to older patients (17% vs. 32%, *p* = 0.018), in pre-menopausal compared to post-menopausal patients (8% vs. 28%, *p* = 0.010), and in those receiving LHRH therapy compared to those who did not (9% vs. 26%, *p* = 0.080). Drug discontinuation was less frequently observed in patients with ECOG PS 0/1 compared to those with ECOG PS 2/3 (1% vs. 11%, *p* = 0.028).

Multivariate analysis revealed that ribociclib was significantly associated with a lower incidence of grade ≥ 3 hematologic toxicity (OR: 0.37, 95% CI: 0.20–0.70, *p* = 0.002; Appendix A). Similarly, ribociclib use was linked to a lower incidence of grade ≥ 3 neutropenia (OR: 0.41, 95% CI: 0.22–0.76, *p* = 0.004; Table 4). In the analysis of treatment interruptions, ribociclib use was associated with fewer interruptions (OR: 0.30, 95% CI: 0.16–0.58, *p* < 0.001; Table 5). Finally, the multivariate analysis for dose reduction identified ribociclib use (OR: 0.37, 95% CI: 0.18–0.76, *p* = 0.007) and pre-menopausal status (OR: 0.17, 95% CI: 0.04–0.74, *p* = 0.018) as factors associated with fewer dose reductions (Table 6).

## 4. Discussion

In this study, we demonstrated that the frequency of grade ≥ 3 hematologic toxicities, including anemia, neutropenia, and thrombocytopenia, did not increase in HR+ HER2-negative metastatic breast cancer patients receiving CDK4/6 inhibitors in combination with palliative radiotherapy compared to those not receiving radiotherapy. An inverse relationship was observed between the use of ribociclib and the incidence of grade ≥ 3 hematologic toxicity, grade ≥ 3 neutropenia, drug interruptions, and drug withdrawal. Additionally, menopausal status was identified as a significant factor associated with the likelihood of dose reduction.

CDK4/6 inhibitors are frequently associated with grade ≥ 3 hematologic adverse events. Bone metastases are common in patients with HR+ HER2-negative metastatic breast cancer treated with these agents. In cases of bone metastases, palliative radiotherapy may be necessary to prevent fractures and manage pain. Therefore, it is crucial to understand the impact of palliative radiotherapy on the incidence of hematologic adverse events in patients undergoing treatment with CDK4/6 inhibitors.

The efficacy of concurrent or sequential use of CDK4/6 inhibitors and radiotherapy is currently being evaluated in both preclinical and clinical studies, including the NCT03691493 (ASPIRE) and NCT03870919 (PALATINE) trials [13]. The most significant adverse effects of both treatment modalities are hematologic. Our study examined the impact of concurrently using these two modalities, which have overlapping side effects, on the incidence of hematologic and non-hematologic toxicities.

Ionizing radiation induces double-strand breaks and single-strand lesions in DNA, leading to significant genomic instability and potential cell death [14,15]. When DNA sustains irreparable damage, a signaling cascade involving critical proteins like ATM, Chk2, and p53 is triggered. This cascade results in the cell cycle arrest at the G1/S phase, providing an opportunity for DNA repair or, if the damage is too severe, initiating apoptosis [16]. When single-strand breaks occur, the cell cycle is arrested at the G2/M phase via a pathway involving ATR and Chk1. In cancer cells, which often lose p53 function, the ability to halt the cell cycle at the G1/S phase is compromised, leaving the G2/M phase as the primary checkpoint for arrest [17]. Cyclin-dependent kinases (CDKs) are serine/threonine protein kinases that regulate cell cycle progression by interacting with cyclin D, retinoblastoma protein (Rb), and E2F. CDK4/6 inhibitors act by binding to the ATP-binding domain of CDK4/6, but they differ in their specific cyclin-D targets. Ribociclib primarily targets D1-CDK4 and D3-CDK6, and palbociclib inhibits D1-CDK4, D2-CDK6, and D3-CDK4, while abemaciclib targets D1-CDK4 and D1-CDK6 [18]. CDK4/6 inhibitors impede cell cycle progression by inhibiting the phosphorylation of the Rb protein, effectively arresting the cell cycle at the G1/S phase transition [19,20]. CDK4/6 inhibitors are believed to potentially diminish the effectiveness of radiotherapy by arresting the cell cycle at the G1/S phase, thereby preventing cells from progressing to more radiosensitive phases [21]. The ideal treatment combination and timing to achieve synergistic effects of these two treatment modalities are not yet clear.

In our study, during the 18-month follow-up period, the median PFS and OS were not reached, which is consistent with the literature [1,22,23,24]. Our study included patients who were using palbociclib or ribociclib as CDK4/6 inhibitors. Abemaciclib was not included due to the lack of reimbursement in our country’s healthcare system.

Our study observed that the rates of grade ≥ 3 hematologic toxicities, including anemia, neutropenia, and thrombocytopenia, did not increase in patients receiving CDK4/6 inhibitors who also underwent palliative radiotherapy compared to those who did not receive radiotherapy. Most studies that report contrasting results are retrospective and involve small patient cohorts. Our findings align with the meta-analysis conducted by Kubeczko and colleagues, which included data from 1133 patients [25].

In our study, grade ≥ 3 neutropenia was observed in 47% of patients. This is lower than the incidences reported in clinical trials for ribociclib, where grade ≥ 3 neutropenia was seen in 64% of patients in both the MONALEESA-2 [2] and MONALEESA-7 [26] trials and 54% in the MONALEESA-3 trial [23]. Similarly, palbociclib trials reported higher rates of grade ≥ 3 neutropenia, with 54% in PALOMA-4 [27], 70% in PALOMA-3 [24], and 66% in PALOMA-2 [1].

The lower incidence in our study may be explained by the younger age of our patient cohort, their lower tumor burden, and fewer instances of extensive bone metastases. The rates of grade ≥ 3 anemia and thrombocytopenia in our study were consistent with those reported in the literature. Additionally, while 23% of patients experienced elevated ALT levels, only 1% had grade ≥ 3 ALT elevation.

In our study, treatment interruption, dose reduction, and discontinuation rates of CDK4/6 inhibitors were 55%, 24%, and 2%, respectively. These findings are consistent with those reported in the PALOMA-2 [1] study, which observed interruption and dose reduction rates of 67% and 36%, and the MONALEESA-3 [23] trial, which reported a 38% dose reduction rate. The similarity between our results and previous safety data reinforces the consistency of these outcomes. Furthermore, the interruption and dose reduction rates were similar between patients who received radiotherapy and those who did not. The rate of CDK4/6 inhibitor discontinuation in our study was also in line with that in the established literature.

Kubeczko et al. conducted a study investigating the safety of both concurrent and sequential radiotherapy [28]. In this study, the frequency of early-onset neutropenia after radiotherapy in CDK4/6 inhibitor users was similar between those who received radiotherapy and those who did not. However, the incidence of neutropenia increased in patients who received radiotherapy starting from the second half of the second cycle. Our study did not evaluate the timing between radiotherapy and grade ≥ 3 hematologic toxicity onset. Additionally, the dose reduction rate was consistent between patients who received radiotherapy and those who did not.

Our study revealed that ribociclib use was associated with lower incidences of grade ≥ 3 hematologic toxicity, grade ≥ 3 neutropenia, drug interruption, and drug discontinuation compared to palbociclib. Although these findings are consistent with those reported in clinical trials, it is important to highlight that no studies have directly compared these two drugs head to head [29].

The primary limitation of our study is its retrospective nature. Furthermore, we did not assess the interval between radiotherapy and the onset of grade ≥ 3 hematological adverse events. Despite these challenges, our study stands out as one of the most extensive single-center experiences reported to date.

## 5. Conclusions

In conclusion, the administration of concomitant palliative radiotherapy during CDK4/6 inhibitor therapy does not elevate the risk of grade ≥ 3 hematologic adverse events. Moreover, ribociclib is associated with a lower incidence of grade ≥ 3 hematologic toxicity, grade ≥ 3 neutropenia, drug interruptions, and dose reductions, suggesting a more favorable safety profile than other CDK4/6 inhibitors.

## Figures and Tables

**Table 1 cancers-17-00424-t001:** Tumor and patient characteristics and adverse events.

Age, median (IQR)		57.5 (46–68)
Gender (female)		185 (98%)
Menopause	Pre-	38 (20%)
	Post-	148 (79%)
ECOG	0	6 (3%)
	1	163 (86%)
	2	14 (8%)
	3	5 (3%)
HER2	Score 0	143 (79%)
	Score 1	28 (15%)
	Score 2, FISH: negative	11 (6%)
Metastatic area	Bone	124 (66%)
	Liver	44 (23%)
	Lung	22 (12%)
	Lymph node	42 (22%)
	Skin	12 (7%)
Extensive bone metastases		87 (46%)
Vertebral bone metastases		89 (47%)
Received before chemotherapy		72 (37%)
Chemotherapy within one year before CDK4/6		23 (12%)
De novo metastatic		119 (63%)
Line	First	143 (76%)
	Second	41 (22%)
	Third and after	4 (2%)
CDK4/6 inhibitors	Palbociclib	71 (38%)
	Ribociclib	117 (62%)
Co-administered therapy	AI	113 (60%)
	AI + LHRH	18 (10%)
	Fulvestrant	52 (28%)
	Fulvestrant + LHRH	4 (2%)
Concurrent radiotherapy		47 (25%)
Radiotherapy Site		
Spine		29 (60%)
Femur		9 (19%)
Pelvis		4 (9%)
Sternum		2 (4%)
Humerus		2 (4%)
Skull		1 (2%)
(All Grades) hematologic toxicity		173 (92%)
(All Grades) anemia		122 (65%)
(All Grades) neutropenia		169 (90%)
(All Grades) thrombocytopenia		54 (29%)
Grade ≥ 3 hematologic toxicity		97 (52%)
Grade ≥ 3 anemia		19 (10%)
Grade ≥ 3 neutropenia		89 (47%)
Grade ≥ 3 thrombocytopenia		13 (7%)
All grades of acute kidney injury		6 (3%)
All grades of ALT increase		43 (23%)
Grade ≥ 3 ALT increase		1 (1%)
All grades of QT prolongation		2 (1%)
Grade ≥ 3 dermatitis		1 (1%)
CDK4/6 inhibitors	Interruption	104 (55%)
	Dose reduction	45 (24%)
	Withdrawal	3 (2%)
Progression		62 (33%)
Death		35 (19%)
Follow-up, months		18.5 (0.1–49.5)
PFS, median, IQR		NE
OS, median, IQR		NE

The table provides a detailed overview of patient demographics, tumor features, treatment regimens, and the incidence of various adverse events observed during the study. Key characteristics include median age, ECOG performance status, hormone receptor status, metastatic sites, prior chemotherapy use, and the administration of CDK4/6 inhibitors. The table also summarizes the rates of hematologic toxicities, both overall and grade ≥ 3, as well as information on treatment interruptions, dose reductions, and patient outcomes. ECOG: Eastern Cooperative Oncology Group (Performance Status); HER2: Human Epidermal Growth Factor Receptor 2; FISH: Fluorescence In Situ Hybridization; CDK4/6: cyclin-dependent kinases 4 and 6; AI: aromatase inhibitor; LHRH: Luteinizing Hormone-Releasing Hormone; ALT: Alanine Aminotransferase; PFS: progression-free survival; OS: overall survival; NE: Not Evaluated; IQR: interquartile range.

**Table 2 cancers-17-00424-t002:** Frequency of hematological and liver toxicity.

	Grade ≥ 3 Hematologic Toxicity (%)	*p*	Grade ≥ 3 Anemia (%)	*p*	Grade ≥ 3 Neutropenia (%)	*p*	Grade ≥ 3 Thrombocytopenia (%)	*p*	All Grades of ALT Increase (%)	*p*
ECOG 0/1 vs. 2/3	52 vs. 47	0.677	10 vs. 11	1.000	48 vs. 47	0.985	7 vs. 11	0.627	24 vs. 11	0.252
AI vs. fulvestrant	48 vs. 59	0.174	8 vs. 14	0.222	43 vs. 57	0.071	5 vs. 11	0.214	21 vs. 27	0.421
LHRH vs. not	46 vs. 52	0.557	5 vs. 11	0.705	46 vs. 47	0.872	5 vs. 7	1.000	14 vs. 24	0.267
Pre- vs. post-menopause	41 vs. 55	0.105	3 vs. 12	0.129	41 vs. 50	0.303	3 vs. 8	0.471	14 vs. 24	0.157
Age (<60 vs. ≥60)	46 vs. 58	0.097	10 vs. 11	0.804	42 vs. 54	0.124	7 vs. 7	0.912	25 vs. 20	0.440
Bone metastasis	51 vs. 53	0.763	10 vs. 11	0.786	46 vs. 50	0.600	7 vs. 8	0.766	24 vs. 20	0.548
Extensive bone metastasis	53 vs. 51	0.745	13 vs. 8	0.284	46 vs. 49	0.728	7 vs. 7	0.993	30 vs. 17	0.034
De novo or not	50 vs. 55	0.468	11 vs. 9	0.625	45 vs. 52	0.312	6 vs. 9	0.554	24 vs. 22	0.778
Palbociclib vs. ribociclib	68 vs. 42	<0.001	16 vs. 7	0.056	61 vs. 39	0.005	10 vs. 5	0.244	23 vs. 23	0.932
First line vs. other	47 vs. 67	0.020	8 vs. 16	0.167	43 vs. 60	0.051	6 vs. 11	0.308	22 vs. 24	0.773
Received chemotherapy before or not	48 vs. 53	0.677	13 vs. 10	0.711	44 vs. 48	0.844	9 vs. 7	0.667	17 vs. 24	0.516
Combined treatment vs. CDK4/6 only	53 vs. 51	0.800	9 vs. 11	0.786	51 vs. 46	0.555	9 vs. 6	0.740	30 vs. 21	0.192

The table compares the incidence of grade ≥ 3 hematologic toxicities, including anemia, neutropenia, and thrombocytopenia, as well as all-grade ALT increases, across different variables such as ECOG performance status, type of endocrine therapy (AI vs. fulvestrant), use of LHRH, menopausal status, age, presence of bone metastases, extensive bone metastases, de novo metastasis, type of CDK4/6 inhibitor (palbociclib vs. ribociclib), line of therapy (first vs. subsequent), prior chemotherapy, and concurrent radiotherapy. Statistical significance (*p*-values) is also provided for each comparison. ECOG: Eastern Cooperative Oncology Group (Performance Status); AI: aromatase inhibitor; LHRH: Luteinizing Hormone-Releasing Hormone; ALT: Alanine Aminotransferase; RT: radiotherapy.

**Table 3 cancers-17-00424-t003:** Frequency of interruption and dose reduction for CDK4/6 inhibitors.

	Interruption (%)	*p*	Dose Reduction (%)	*p*	Withdrawal (%)	*p*
ECOG 0/1 vs. 2/3	57 vs. 42	0.209	22 vs. 37	0.164	1 vs. 11	0.028
AI vs. fulvestrant	51 vs. 64	0.098	21 vs. 30	0.188	2 vs. 0	0.555
LHRH vs. not	50 vs. 56	0.610	9 vs. 26	0.080	0 vs. 2	1.000
Pre- vs. post-menopause	46 vs. 59	0.159	8 vs. 28	0.010	0 vs. 2	1.000
Age (<60 vs. ≥60)	48 vs. 64	0.026	17 vs. 32	0.018	1 vs. 2	0.587
De novo or not	52 vs. 61	0.244	23 vs. 26	0.599	3 vs. 0	0.299
Palbociclib vs. ribociclib	73 vs. 44	<0.001	37 vs. 16	0.001	0 vs. 3	0.291
First line vs. other	52 vs. 67	0.079	23 vs. 27	0.623	2 vs. 0	1.000
Received chemotherapy before or not	48 vs. 57	0.418	9 vs. 27	0.062	0 vs. 2	1.000
Combined treatment vs. CDK4/6 only	53 vs. 56	0.735	28 vs. 23	0.490	1 vs. 2	1.000

The table presents a comparison of these outcomes based on ECOG performance status, type of endocrine therapy (AI vs. fulvestrant), use of LHRH, menopausal status, age, presence of de novo metastasis, type of CDK4/6 inhibitor (palbociclib vs. ribociclib), line of therapy (first vs. subsequent), prior chemotherapy, and concurrent radiotherapy. Statistical significance (*p*-values) is provided for each comparison. ECOG: Eastern Cooperative Oncology Group (Performance Status); AI: aromatase inhibitor; LHRH: Luteinizing Hormone-Releasing Hormone; RT: radiotherapy.

**Table 4 cancers-17-00424-t004:** Univariate and multivariate analysis results on grade ≥ 3 neutropenia.

	Univariate Analysis			Multivariate Analysis		
	OR	95% CI	*p*	OR	95% CI	*p*
AI (AI vs. fulvestrant)	0.56	0.30–1.05	0.072	0.55	0.29–1.06	0.074
Age (<60 vs. ≥60)	0.64	0.36–1.13	0.125			
Ribociclib (ribociclib vs. palbociclib)	0.42	0.23–0.77	0.005	0.41	0.22–0.76	0.004
First line vs. other	0.51	0.26–1.01	0.053			

The table displays the odds ratios (ORs), 95% confidence intervals (CIs), and *p*-values for factors influencing the risk of grade ≥ 3 neutropenia, including type of endocrine therapy (AI vs. fulvestrant), age, CDK4/6 inhibitor used (ribociclib vs. palbociclib), and line of therapy (first line vs. others). The results are presented for both univariate and multivariate analyses to identify independent predictors of neutropenia. OR: odds ratio; CI: confidence interval; AI: aromatase inhibitor.

**Table 5 cancers-17-00424-t005:** Univariate and multivariate analysis results on interruption.

	Univariate Analysis			Multivariate Analysis		
	OR	95% CI	*p*	OR	95% CI	*p*
AI (AI vs. fulvestrant)	0.58	0.31–1.10	0.100	0.54	0.27–1.06	0.073
Age (<60 vs. ≥60)	0.51	0.29–0.93	0.027	0.55	0.29–1.02	0.057
Ribociclib (ribociclib vs. palbociclib)	0.29	0.15–0.55	<0.001	0.30	0.16–0.58	<0.001
First line vs. other	0.54	0.27–1.08	0.082			

The table presents the odds ratios (ORs), 95% confidence intervals (CIs), and *p*-values for factors influencing the likelihood of treatment interruption, including type of endocrine therapy (AI vs. fulvestrant), age, CDK4/6 inhibitor used (ribociclib vs. palbociclib), and line of therapy (first-line vs. others). Both univariate and multivariate analyses are shown to identify independent predictors of treatment interruption. OR: odds ratio; CI: confidence interval; AI: aromatase inhibitor.

**Table 6 cancers-17-00424-t006:** Univariate and multivariate analysis results on dose reduction.

	Univariate Analysis			Multivariate Analysis		
	OR	95% CI	*p*	OR	95% CI	*p*
AI (AI vs. fulvestrant)	0.62	0.31–1.26	0.190			
Received LHRH vs. not	0.28	0.06–1.27	0.099			
Age (<60 vs. ≥60)	0.49	0.18–1.34	0.165			
Ribociclib (ribociclib vs. palbociclib)	0.34	0.17–0.67	0.002	0.37	0.18–0.76	0.007
Pre-menopause vs. post-menopause	0.22	0.07–0.76	0.017	0.17	0.04–0.74	0.018
ECOG 0/1 vs. 2/3	0.49	0.18–1.34	0.165			
Received chemotherapy before or not	0.26	0.06–1.17	0.080			

The table displays the odds ratios (ORs), 95% confidence intervals (CIs), and *p*-values for factors associated with the likelihood of dose reduction, including type of endocrine therapy (AI vs. fulvestrant), use of LHRH, age, CDK4/6 inhibitor used (ribociclib vs. palbociclib), menopausal status (pre-menopause vs. post-menopause), ECOG performance status, and prior chemotherapy. Both univariate and multivariate analyses are presented to identify independent predictors of dose reduction. OR: odds ratio; CI: confidence interval; AI: aromatase inhibitor; LHRH: Luteinizing Hormone-Releasing Hormone; ECOG: Eastern Cooperative Oncology Group (Performance Status).

## Data Availability

The data supporting the findings of this study are available from the corresponding author upon reasonable request. In adherence to ethical standards and institutional policies, only fully anonymized datasets will be shared to protect patient confidentiality. Data access will be granted exclusively for legitimate academic and non-commercial research purposes, following a formal request and subject to approval in line with institutional and ethical regulations.

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
