# Peer review of "Palliative Radiotherapy in Metastatic Breast Cancer Patients on CDK4/6 Inhibitors: Safety Analysis"

_cancers, 2025, doi:10.3390/cancers17030424_

Round 1
Reviewer 1 Report
Comments and Suggestions for Authors
1
1. In Line 113 –Result section is mentioned that 47 patients received 20Gy/5 fractions. I recommend to be mentioned the fractionation schemas used in this study.
It would benefit to investigate if there are correlations between fractionation schemas 20Gy/5fr or 30Gy/10fr and hematological toxicity grade 3 or more.
. Concomitant radiotherapy was administrated at 25% of patients.
I recommend would be a benefit to be presented: how many CD4/6 inhibitors cycles have been administrated for patients who benefited by concurrent radiotherapy. Also, it would be a benefit to investigate if there are correlations between number cycles of CDK4/6 inhibitors with hematological toxicity, grade 3 or more.
3. I recommend to be mentioned if patients from study performed radiotherapy on one or more than one bone metastatic sites.
Author Response
We would like to express our sincere gratitude to Reviewer 1 for their detailed and insightful comments, which have greatly contributed to improving the quality and clarity of our manuscript. Your thoughtful suggestions and critical observations have allowed us to address key aspects of our study and enhance its overall impact. We deeply value the time and effort you have dedicated to reviewing our work, and we have carefully considered each of your recommendations.
Reviewer-1
In Line 113 –Result section is mentioned that 47 patients received 20Gy/5 fractions. I recommend to be mentioned the fractionation schemas used in this study.
Authors’ Response: Thank you for this valuable suggestion. We have revised the manuscript to include a detailed description of the radiotherapy fractionation schemas employed in this study. Specifically, 44 patients received a regimen of 20Gy in 5 fractions, while 3 patients were treated with 30Gy in 10 fractions. The selection of fractionation and dose was determined by the treating clinician, guided by established palliative radiotherapy recommendations (PMID: 27663933). Higher doses, such as 30Gy, were typically applied to regions like the sternum and lumbar vertebrae.
It would benefit to investigate if there are correlations between fractionation schemas 20Gy/5fr or 30Gy/10fr and hematological toxicity grade 3 or more.
Authors’ Response: Thank you for this insightful suggestion. We agree that exploring correlations between fractionation schemas and grade ≥3 hematological toxicity would provide valuable additional information. However, in our study, only three patients received the 30Gy/10fr regimen, which is insufficient for a statistically meaningful comparison with the 20Gy/5fr group. The small sample size in this subgroup limits the ability to draw robust conclusions regarding potential differences in hematological toxicity between these fractionation schemas.
Concomitant radiotherapy was administrated at 25% of patients.
I recommend would be a benefit to be presented: how many CD4/6 inhibitors cycles have been administrated for patients who benefited by concurrent radiotherapy. Also, it would be a benefit to investigate if there are correlations between number cycles of CDK4/6 inhibitors with hematological toxicity, grade 3 or more.
Authors’ Response: Thank you for this valuable comment. We have provided additional details regarding the number of CDK4/6 inhibitor cycles administered to patients who received concomitant radiotherapy. Specifically, 32 patients received radiotherapy within the first 3 months of CDK4/6 inhibitor treatment, 6 patients received radiotherapy during the second 3 months, and 9 patients received radiotherapy after 6 months of treatment. While we acknowledge the importance of investigating potential correlations between the number of CDK4/6 inhibitor cycles and grade ≥3 hematological toxicity, the relatively small number of patients in certain subgroups limits the statistical power of such an analysis.
I recommend to be mentioned if patients from study performed radiotherapy on one or more than one bone metastatic sites.
Authors’ Response: Thank you for your thoughtful comment. In our study, palliative radiotherapy was administered to a single metastatic bone site per patient. The decision to limit radiotherapy to one site was based on clinical necessity, prioritizing the most symptomatic or high-risk lesion requiring treatment. This approach aligns with standard palliative care practices, where the goal is to provide effective symptom management and reduce the risk of complications such as fractures. We have clarified this in the manuscript to ensure greater transparency regarding the radiotherapy protocols employed in the study.
Once again, we would like to thank you for your valuable comments and constructive feedback. We have carefully addressed all the points you raised and incorporated the suggested changes into the manuscript. Specifically, we have clarified the fractionation schemas, included details about radiotherapy timing and its correlation with CDK4/6 inhibitor use, and specified that radiotherapy was limited to a single metastatic bone site per patient. These revisions have been made in the relevant sections of the manuscript, as outlined in our detailed responses.
We are confident that these changes have strengthened the manuscript, and we hope it now meets your expectations.

Reviewer 2 Report
Comments and Suggestions for Authors
In this analysis the authors report on a cohort of 188 metastatic breast cancer patients who received CDK4/6 inhibitors combined – in 25% of the cases – with concurrent radiotherapy. The topic is of interest to the scientific community since the combination of radiotherapy with targeted drugs is an emerging field of studies which should provide answers both to toxicity questions as well as clinical outcome. The authors conclude that radiotherapy (RT) and CDK4/6 inhibitors can be combined without excess toxicity.
1. From the way the data are presented, I doubt whether this conclusion can be drawn easily. Hence I would recommend major changes with respect to how the information in the current manuscript is organized. Twenty-five percent of the patients received combined treatment therefore it seems logical to me that this group is compared to the rest of the patients. Having said this, tables should be re-organized accordingly (combined treatment versus CDK4/6 only). Only if there is no statistical difference between groups the conclusion drawn by the authors (lines 308-310) is correct.
2. Line 104: PFS and OS rates are briefly touched on. The authors might want to present the respective Kaplan-Meier curves in the manuscript preferably comparing the combined group to the CDK4/6 group only (log-rank test).
3. Lines 111-115:
· How many of the patients received RT? 47 or 46?
· What is meant by calvarium? Is it the skull? Also in table 1
4. Radiotherapy: Why exactly did you choose 20 Gy in 5 fractions? Did you use this regimen in all the the mentioned locations?
Comments on the Quality of English Language
There are no major language concerns.
Author Response
We extend our heartfelt gratitude to Reviewer 2 for their thorough review and insightful comments, which have greatly contributed to improving the quality and clarity of our manuscript. Your thoughtful observations have highlighted important areas for refinement, and we have carefully addressed each of your suggestions. We deeply value your feedback and the opportunity to enhance the scientific rigor of our work.
In this analysis the authors report on a cohort of 188 metastatic breast cancer patients who received CDK4/6 inhibitors combined – in 25% of the cases – with concurrent radiotherapy. The topic is of interest to the scientific community since the combination of radiotherapy with targeted drugs is an emerging field of studies which should provide answers both to toxicity questions as well as clinical outcome. The authors conclude that radiotherapy (RT) and CDK4/6 inhibitors can be combined without excess toxicity.
- From the way the data are presented, I doubt whether this conclusion can be drawn easily. Hence I would recommend major changes with respect to how the information in the current manuscript is organized. Twenty-five percent of the patients received combined treatment therefore it seems logical to me that this group is compared to the rest of the patients. Having said this, tables should be re-organized accordingly (combined treatment versus CDK4/6 only). Only if there is no statistical difference between groups the conclusion drawn by the authors (lines 308-310) is correct.
Thank you for your insightful comment and for emphasizing the importance of a clear and logical organization of the data. We fully agree with your suggestion to ensure that the group of patients receiving combined treatment (palliative radiotherapy with CDK4/6 inhibitors) is appropriately compared to those receiving CDK4/6 inhibitors alone. To address this, we have reorganized the tables to present the data as a comparison between the combined treatment group and the CDK4/6-only group. In Supplemental Table 2, we have provided detailed patient characteristics for both groups, noting that most characteristics were similar between them. However, we observed that the rate of de novo metastases and the use of CDK4/6 inhibitors as first-line treatment were higher in the group that did not receive radiotherapy. Additionally, we have reviewed the statistical analyses and confirmed that no significant differences in outcomes were observed between the two groups. This supports the conclusion that the use of palliative radiotherapy during CDK4/6 inhibitor therapy does not increase the risk of grade ≥3 hematologic toxicity.
These points have been addressed in the main text.
- Line 104: PFS and OS rates are briefly touched on. The authors might want to present the respective Kaplan-Meier curves in the manuscript preferably comparing the combined group to the CDK4/6 group only (log-rank test).
Thank you for your valuable suggestion. Kaplan-Meier survival curves illustrating progression-free survival (PFS) and overall survival (OS) have been added to the manuscript. These curves compare patients receiving combined treatment (palliative radiotherapy with CDK4/6 inhibitors) to those receiving CDK4/6 inhibitors alone, and statistical differences were evaluated using the log-rank test.
- Lines 111-115:
How many of the patients received RT? 47 or 46?
Thank you for your observation. We confirm that 47 patients received palliative radiotherapy in our study. The discrepancy regarding the number of patients receiving spine radiotherapy has been corrected in the manuscript to reflect this total accurately.
What is meant by calvarium? Is it the skull? Also in table 1
Thank you for pointing out the terminology in our manuscript. You are correct that "calvarium" refers to the skull. To ensure clarity and consistency, we have replaced the term "calvarium" with "skull" in the text and in Table 1.
- Radiotherapy: Why exactly did you choose 20 Gy in 5 fractions? Did you use this regimen in all the the mentioned locations?
Thank you for this important question. The decision to use the 20Gy in 5 fractions regimen was based on recommendations from the ASTRO (American Society for Radiation Oncology) guidelines (PMID: 27663933), which suggest this dose as a standard for palliative radiotherapy in metastatic bone disease. This regimen is effective in providing pain relief and preventing skeletal-related events while minimizing treatment burden for patients.
Although 20Gy in 5 fractions was the most used regimen, adjustments were made in specific cases based on clinical judgment and patient needs. For instance, doses such as 30Gy in 10 fractions were applied to certain locations like the sternum and lumbar vertebrae, where higher palliative benefit was deemed necessary. These variations have been clarified in the manuscript.
Once again, we sincerely thank you for your valuable feedback and constructive suggestions. We have carefully considered and addressed all the points raised, incorporating the recommended changes into the manuscript. These revisions include the clarification of data presentation, the addition of survival analysis details, and addressing methodological considerations to provide a more robust and transparent account of our findings.
We believe these revisions have significantly strengthened the manuscript and hope it now meets your expectations. Thank you for your time and effort in reviewing our work.

Reviewer 3 Report
Comments and Suggestions for Authors
Fortunately, breast cancer survival is increasing every day, mainly due to better early diagnosis.
New treatments for metastatic cancer have been incorporated that have improved survival and disease-free survival in these patients with advanced disease and poor prognosis.
The MONALEESA-2, MONALEESA-3, and MONALEESA-7 studies have positioned the role of CD4 inhibitors in these ER positive and HER2 negative patients, compared to hormone therapy, together with palliative radiotherapy, but with a high rate of side effects.
The present study sheds new light on the use of these treatments, showing acceptable hematological side effects for patients.
I congratulate the authors for this work. The number of patients evaluated in this article is very adequate. However, a prospective randomized study would be desirable, but fortunately the number of patients with these characteristics would require a multicentric study to recruit patients. I encourage authors to lead this type of work.
Author Response
Fortunately, breast cancer survival is increasing every day, mainly due to better early diagnosis.
New treatments for metastatic cancer have been incorporated that have improved survival and disease-free survival in these patients with advanced disease and poor prognosis.
The MONALEESA-2, MONALEESA-3, and MONALEESA-7 studies have positioned the role of CD4 inhibitors in these ER positive and HER2 negative patients, compared to hormone therapy, together with palliative radiotherapy, but with a high rate of side effects.
The present study sheds new light on the use of these treatments, showing acceptable hematological side effects for patients.
I congratulate the authors for this work. The number of patients evaluated in this article is very adequate. However, a prospective randomized study would be desirable, but fortunately the number of patients with these characteristics would require a multicentric study to recruit patients. I encourage authors to lead this type of work.
Thank you for your encouraging and thoughtful feedback. We greatly appreciate your recognition of our work and your acknowledgment of the importance of this study in shedding light on the safety of combining CDK4/6 inhibitors with palliative radiotherapy in patients with ER-positive, HER2-negative metastatic breast cancer.
We agree that while our findings provide valuable real-world insights, a prospective randomized study would indeed offer more robust evidence. As you noted, conducting such a study would require a multicentric approach to ensure sufficient patient recruitment, given the specific characteristics of this patient population. We are inspired by your suggestion and fully recognize the need for such research to further validate and expand upon our results. Moving forward, we will explore opportunities to lead or participate in collaborative multicentric trials to address this critical area of oncology practice.
Once again, thank you for your kind words and constructive suggestions. We are grateful for your thoughtful review and encouragement.

Round 2
Reviewer 2 Report
Comments and Suggestions for Authors
I have no further comments